## [Transparent Peer Review file · Nature Communications]

3D Pentaculture Model Unveils Malignant Cell-Driven Macrophage Polarization in High-Grade Serous Ovarian Cancer

Corresponding Author: Professor Frances (R) Balkwill

Version 0:

Reviewer comments:

Reviewer #1

(Remarks to the Author)

Malacrida et al. present a pentaculture model incorporating HGSOV cancer cell lines alongside primary environmental cells, including donor-derived monocytes, to analyze their impact. The robustness of the study is a concern, as key experimental controls are often missing and statistical support is inconsistently applied. Several claims, particularly those based on bulk RNA sequencing, appear overinterpreted given the available data. Furthermore, it is unclear how the presented model meaningfully advances beyond the authors' previous work or compares to more advanced organoid-based systems that already incorporate multicellular microenvironments (e.g., Polak et al., *Nature Reviews Cancer*, 2024). The investigation into the influence of cancer cells on macrophages is not well executed and lacks sufficient contextualization within the existing literature or in comparison to other *in vitro* models. Similarly, the exploration of CD47 and CD24 is not new and lacks sufficient referencing. While the proposed alternative resistance mechanisms could represent a more compelling direction, they remain preliminary and are not supported by clear mechanistic insight. Overall, the structure of the manuscript could be revisited and streamlined, particularly in how the model is characterized and contextualized relative to existing *in vitro* systems. See further comments below.

- Although the pentaculture is maintained for 14 days, most analyses are conducted at day 7. If day 14 is not a relevant time point due to suboptimal culture conditions, it should be omitted; if it is relevant, it warrants proper analysis. Overall, the choice of the most appropriate time point should be clearly justified and supported by data. For example, what are the proportions of the different cell types, and how do these change over the course of the culture? The authors should include a brightfield or other microscopy-based time course over the 14-day period, along with measurements of cell viability, proliferation, and cell death.

- Why did the authors choose to use cancer cell lines rather than primary tumor cells? Using primary material could provide greater physiological relevance.

- The authors conclude from bulk RNA sequencing that unsupervised clustering segregates samples according to malignant cell types rather than the donor of the normal cells, suggesting that malignant cells are the primary drivers (Figure 2B). However, this conclusion is not supported by this presented analysis. Bulk RNA-seq inherently masks cellular heterogeneity, making it insufficient to attribute clustering solely to malignant cell types. Moreover, what stands out more clearly is that the Pentacultures cluster together as a group, and not with their corresponding cancer cell types. If the authors aim to explore intercellular influences or the contribution of specific cell types within these cultures, single-cell RNA sequencing would be a more appropriate and informative approach.

In Figure 2C, how can the authors conclude that differentiation into macrophages occurred in their culture based solely on GO term analysis? These GO terms are also enriched in monocytes, so this does not provide definitive evidence for macrophage differentiation.

In Figure 2D, it is unclear which GO terms specifically support the conclusion of a tissue-like environment. The figure is poorly presented and does not convincingly support the authors' claims. Simply listing GO terms feels more like a supplementary figure. Higher-resolution data, with single-cell RNA sequencing, along with validation of key marker genes at

the protein level, with appropriate controls, would be necessary to robustly substantiate the findings.

- It is unclear whether the genomic and transcriptomic profiling of the cancer cell lines was performed on the isolated cell lines or within the context of the Pentaculture system. This distinction is important, as the microenvironment can significantly influence cancer gene expression. Additionally, this characterization of the cancer cell lines feels disconnected from the main narrative of the manuscript. The conclusions drawn regarding their relevance to macrophage states are underdeveloped and lack clear integration into the broader context of the study. Also, it is broadly known in many cancer and literature on this topics should be thoroughly reviewed by the authors (e.g. Weverwijk, De visser, nature review cancer 2023)

- Statistical analyses are missing from many figure panels, but their inclusion would be important to appropriately support and nuance the authors' claims.

-Legends need more information

- Fig 1 D axis labeling and legend not clear.

- The graphical presentation in Fig. 1F is somewhat unclear. Were multiple donors analyzed? Additionally, could you include data for day 0 (monocytes) and day 14?

- In Fig3, I-J, I am missing the monocytes as control.

- Although the video quality is high, the objective and key observations are not clearly communicated. Adding annotations and labels would greatly enhance clarity by guiding the viewer to the relevant events. Specifically, it is not evident that CD47 blockade results in increased phagocytosis. If this is a key conclusion, more direct visual evidence of phagocytosis should be shown. Additionally, the significance of the observed wobbling motion is unclear, why should this be interpreted as an indicator of phagocytic activity? Clarification or supporting rationale would be helpful.

-The conclusion that increased phagocytic activity in post-chemotherapy samples from responders, compared to those from relapsed patients, implies that enhancing macrophage phagocytic capacity may improve chemotherapy response is not well supported. An alternative explanation could be that effective chemotherapy leads to greater tumor cell death and debris accumulation, which in turn triggers macrophage activation and phagocytic activity as a secondary effect, rather than a cause of therapeutic success. Could the authors clarify whether their data distinguish between these possibilities, and comment on the directionality of this relationship?

Reviewer #2

(Remarks to the Author)

Malacrida et al. report the development and initial characterization of high grade serous ovarian cancer (HGSO) "pentacultures". This organotypic cell system includes cancer cells, mesothelial cells, fibroblasts, adipocytes and monocytes, the latter of which differentiate into macrophages in the culture. The adipocytes, mesothelial cells, and fibroblasts are obtained from omental biopsies and cultured in vitro. Three separate donors are used for this study, although the authors imply that they have studied many more (see below). The monocytes are purified from peripheral blood of a single donor. Using these pentacultures, the authors report several key findings. First, tumor cells seem to drive most of the macrophage polarization in the cultures, as shown by differences in transcriptomic and proteomic data, which had more effect than varying the origin (i.e. donors) of the other cell types. Second, each HGSO cell line drives different degrees of macrophage polarization and different spatial architecture. Third, the authors report intriguing macrophage-directed effects on tumour cell viability, which could, at least for one cell line, be modulated with monoclonal antibodies targeting "do not eat me" signals.

Overall, this is an interesting, well written, and technically well-performed study that should be of general interest to the readership of Nature Communications. There are, however, some conceptual and technical issues that should be addressed in a revision.

Major Issues:

(1) A penta-culture (actually, a sexta-culture, as it includes endothelium) of the ovarian omentum, incorporating a macrophage cell line, has been described previously (Estermann et al., Biomaterials, 2023), albeit that work was more focused on dissemination of HGSO within the omentum than on the issues discussed here. In some ways, the authors of the current paper have also reported a 5-cell type culture (see Malacrida et al. 2021, Figure 4F), although that work focused on platelets, not macrophages. The authors should discuss the similarities and differences between their system and the previous reports. Also, they should emphasize the novelty of this paper, which is the demonstration that tumour cells drive the TME (at least those components of the TME that are present in this culture), and the "don't eat me" aspects. The utility of their system in enabling the study of monocyte-> macrophage differentiation should also be discussed.

(2) Surprisingly, the authors do not comment on whether the tumor cells also drive fibroblast differentiation to CAFs (or induce additional fibroblast/CAF differentiation from mesothelium). Alternatively, does culturing the primary omental fibroblasts in vitro prior to their incorporation into pentacultures induce activation similar to that of CAFs? Several reports in other tumor systems have emphasized the role of CAFs in directing macrophage polarization, so this point would seem to be

important.

(3) The authors state that they used 3 donors for the omentum-derived cells for this study, but in the Discussion (line 366), they say there are >50 donors for the first three cell types and 5 for the monocytes. To what extent have these other donors been characterized in this system? Also, what types of surgeries were the patients who donated omentum undergoing (i.e., what were their underlying disorders)?

(4) In Figure 1, the authors indicate that monocytes spontaneously differentiate into macrophages when introduced into the quad-culture model. An important missing control would seem to be incubating the monocytes without any cancer cells; have they done this? Also, even if the cancer cells are necessary to start the monocyte differentiation process, are they sufficient? Or is there cross talk with other cell types (e.g., CAFs, as per point 2)?

(5) The authors perform the studies in this paper over 7 days. How long do the cultures last beyond that? Also, they comment (and show data that) the viability of the cancer cells is preserved over the 7-day period, but what about the other cell types?

(6) They note that a major limitation of the current work is the absence of adaptive immune cells (I would include other innate immune cells, e.g., gMDSC/neutrophils, NK cells). A priori, it would seem that at least some of these could be added. Could the authors comment on whether they have tried to do so in the Discussion. Also, is there a reason why they chose to use immortalized HGSOC lines instead of primary tumor cells or organoids?

(7) Line 151-154: the authors state that CIBERSORTx "revealed that fibroblast and mesothelial cells were present at similar abundance..", but to my eyes, it looks like there are significantly more of these cells in AOCS1 cultures—am I wrong?

(8) Lines 159-60, the authors correctly note that the percentage of G164 cells is significantly higher by flow, as predicted by CIBERSORTx. But why are AOCS1 and OVCAR1 identical when they are significantly different in the CIBERSORTx analysis?

(9) Line 197, Why don't the authors also comment on CDK12 loss in this line? And what about CCNE1 amplification in OVCAR3?

(10) Lines 311-313, referring to the G164 pentacultures, the authors state that macrophages resided mainly in the stroma. But from the micrograph in 6D, it looks mainly like there are far fewer macrophages, period. Could they comment?

(11) The quantification in Fig. 5D does not agree with the photo in 5C, at least to my eyes.

Minor issues:

- 1) Line 39, HGSOC is misspelled; Line 306 "the" should be capitalized.
- 2) In Fig 1A, I found it difficult to see the fibroblasts in the cartoon. I suggest using a different color for more contrast.
- 3) Lines 104-106; the authors state that "...pentacultures retained a substantial population of monocytes, as indicated by CD45+ staining..." But macrophages are also CD45+? How can this statement be used to support the retention of MONOCYTES?
- 4) Figs. 2C-E are too small to see without magnification (i.e., when printed out at 1x). So are Fig. 4D and 6A and Supp Figs 2 and 3 (especially those numbers in blue boxes).
- 5) Figure 2F. To facilitate reader interpretation, I recommend adding arrows in the graph that indicate the change of the center of mass of each population between days 3 and 7
- 6) Why aren't all the chemokines and cytokines presented in the same order in Figure 3I for pentaculture and 2D cultures?
- 7) Line 205, CIN should be defined as chromosomal instability when used for the first time.
- 8) Lines 212-214 is worded awkwardly. Also, flow cytometry does not measure "expression"; it measures "surface expression" (e.g., Fig 1 legend, and elsewhere in the paper"

Reviewer #3

(Remarks to the Author)

The authors present an interesting preclinical model for high-grade serous ovarian cancer, incorporating tumor cells, primary fibroblasts, mesothelial cells, adipocytes, and monocytes using three previously established cell lines.

This manuscript builds on their previous work on "tetracultures", further incorporating monocytes from healthy donors to their TME model. They show that the monocytes differentiated into macrophages, without exogenous cytokines, in the pentacultures. The authors then describe the model using IHC, flow cytometry RNAsequencing and whole genome sequencing. They highlighted the potential influence of malignant cell genomic, transcriptomic and proteomic heterogeneity on the TME. The findings on the functional characterization are interesting. The work is highly relevant and timely; however, the strength of the conclusions and translational impact would be improved with more detailed characterisation of the models and source tumors.

1. For the representativeness of the models: Fig 1 B and S Fig 1b: is the human biopsy from the same patient? It is difficult to assess the cell morphology. To conclude how well the models match the human tumors, it would be important to should show 1) matching human tumor and model H&E stainings 2) higher magnification of H&E to allow for assessing the cell morphology 3) It is difficult to see any tumor cells in the models - the authors should show also tumor marker stainings e.g.

CK7, PA8 in the models. Based on Fig1 B and SFig 1B, HLA-DR is widely expressed and most of the cells seem to be macrophages and tumor cells cannot be identified from the figures. A4) are fibroblasts and mesothelial cells present in the tumors and in the cultures? Would be important to see stainings for these cell types as well in both the tumors and in the models.

2. Row 101: "AOCS1 and OvCAR3 cancer cells were located close to the stromal cells in the pentacultures, while G164 cancer cells generally formed clusters away from the stromal cells." Where is this shown? Is this quantified? Deeper characterisation of the full composition (see point 1) would also better support the conclusion that can be made on the influence of the malignant cells on the morphology and composition of the models.

3. For the RNAseq and analysis: Fig 2 a - it is difficult to assess which conditions were sequenced and compared. This figure could be improved to better show the different conditions and comparisons. Figure 2 b is not really informative as part of the results, but rather shows sample-specific clustering. Was there a batch effect? Was that corrected for?

4. Tissue composition: bulkRNAseq deconvolution has known biases for decomposing the signal - the authors should validate these findings using another technology, for example, immunofluorescence or flow cytometry.

5. Fig 3: the tumor cell fractions were quite low after - could this be corrected using higher numbers of cancer cells when establishing the cultures? Did the authors test different ratios of the cells in the setup of the pentacultures?

6. How much of the transcriptional differences of the pentacultures stem from the significantly different gene expression profiles of the tumor cells? To show that the differences in the immune/stromal cell phenotypes are not just reflections of the (often poorly tumor signal - scRNAseq (or bulk from flow-sorted populations) from the pentacultures would significantly strengthen the conclusions.

7. Colour annotations are missing in Figure 4 B. The value of WGS exploration in only three tumor samples remains limited. Is G164 a functionally HRD tumor by e.g. functional RAD51 staining? Exploring more models with HRD vs HRP tumors would be really interesting!

8. In the end, it remains unclear how well do the observed models match the clinical tumors in terms of composition and phenotype. It would strengthen the conclusion if the authors had the original tumor samples, and were able to better show that they recapitulate the source tumors. Further, the conclusion that the tumor genotypes shape the TME would need significantly more models to be established to reach statistical conclusions. E.g. the clinical relevance of these phenotypes could then be expanded to other public clinical datasets for increased impact.

9. Figures 6 G-J only show two datapoints per condition and the statistical comparisons - please add biological replicates to these experiments.

Version 1:

Reviewer comments:

Reviewer #1

(Remarks to the Author)

I have no further comments

Reviewer #2

(Remarks to the Author)

This is a revised version of a manuscript that I reviewed earlier. The authors have made a good faith effort to respond to my queries and suggestions as well as those of the other Reviewers. In my opinion, this paper is quite important and warrants publication. However, I do have a few remaining issues/concerns that should be addressed before publication (not requiring re-review).

1) Most importantly, in my previous review, I asked about the source of the adipocytes, mesothelial cells, and fibroblasts used for the pentacultures. The text of the original manuscript (and this version) states only that they were obtained from patients undergoing "gynecological surgeries." However, in their Response to the Reviewer's Comments', they state that these samples were actually from "uninvolved regions" of omentum in patients undergoing surgery for HGSO. Given that these cells are typically bathed in cancer-associated ascites, and some have received neoadjuvant chemotherapy (!), it is highly likely, that they are not normal. The authors should discuss this possibility and its implications for interpreting their results. Also, I think that they should explicitly mention the source of these cells in the text on p. 4 and indicate whether the various donors used had received neoadjuvant chemotherapy or not.

2) The flow plots in Fig. 1F should be described better—what is the y-axis, for example? Also, why not show the actual values for each population in a bar graph/dot plot or table?

3) In Fig 3B, rather than stating the y-axis measures "% viable cancer cells", it would be better if they stated the exact measurement made either in the figure proper or the legend.

4) The color code in Fig. 4B should be explained.

5) The text of the second paragraph of p. 8 needs editing. It should be "...such as fibroblasts", and in the next sentence, "...and shows why external supplementation.."

6) The reference to Adzibolosu et al. in the legend to Fig. 5 should be number 40, not number 13.

Reviewer #3

(Remarks to the Author)

The authors have addressed my comments effectively.

REVIEWERS COMMENTS

Reviewer #1 (Remarks to the Author):

Malacrida et al. present a pentaculture model incorporating HGSOC cancer cell lines alongside primary environmental cells, including donor-derived monocytes, to analyze their impact. The robustness of the study is a concern, as key experimental controls are often missing and statistical support is inconsistently applied (see Reviewer 1 point 4, for detailed response) Several claims, particularly those based on bulk RNA sequencing, appear overinterpreted given the available data (see Reviewer 1, point 3). Furthermore, it is unclear how the presented model meaningfully advances beyond the authors' previous work or compares to more advanced organoid-based systems that already incorporate multicellular microenvironments (e.g., Polak et al., Nature Reviews Cancer, 2024) (see Reviewer 2, point 1). The investigation into the influence of cancer cells on macrophages is not well executed and lacks sufficient contextualization within the existing literature or in comparison to other in vitro models (see Reviewer 2, point 1). Similarly, the exploration of CD47 and CD24 is not new and lacks sufficient referencing. While the proposed alternative resistance mechanisms could represent a more compelling direction, they remain preliminary and are not supported by clear mechanistic insight. Overall, the structure of the manuscript could be revisited and streamlined, particularly in how the model is characterized and contextualized relative to existing in vitro systems. See further comments below.

1. Although the pentaculture is maintained for 14 days, most analyses are conducted at day 7. If day 14 is not a relevant time point due to suboptimal culture conditions, it should be omitted; if it is relevant, it warrants proper analysis. Overall, the choice of the most appropriate time point should be clearly justified and supported by data. For example, what are the proportions of the different cell types, and how do these change over the course of the culture? The authors should include a brightfield or other microscopy-based time course over the 14-day period, along with measurements of cell viability, proliferation, and cell death.

We appreciate the reviewer's comment regarding the choice of time points. To improve the clarity of our manuscript, we have removed all reference to the 14-day time point. Our analyses focus on day 7 because, as demonstrated by our characterisation experiments, the macrophages within the pentaculture have differentiated by this time point. We have also now included the proportion of the different cell compartments at day 0 and day 7 which reflect the viable cells present in the culture at these time points (Reviewer Figure 1 and new Supplementary Figure 1A).

Reviewer Figure 1 (now Supplementary Figure 1A) - Abundance of the different viable cell populations on the models at day 0 and day 7 using flow cytometry (n=3). Holm-Sidak multiple unpaired t-tests between the different models at day 7 indicated that the models differed significantly in their cancer cell percentages: AOCS1 penta vs.

G164 penta ($p=0.05$) and G164 penta vs. OvCAR3 penta ($p=0.0029$) confirming other data in the original manuscript.

2. Why did the authors choose to use cancer cell lines rather than primary tumor cells? Using primary material could provide greater physiological relevance.

We agree with the reviewer that incorporating primary tumour cells could increase the relevance of our model. However, AOCS1 and G164 were derived and established as primary malignant cell lines, without genetic modification or introduction of viral oncogenes. Although the OvCAR3 cell line was purchased from ATCC, in the manuscript we have extensively characterised all three cell lines at the proteomic, transcriptomic and genomic level. We show that our malignant cells are relevant to the human disease in a similar manner to, for instance, HGSOC organoids. As shown in **Figure 4E**, the three cell lines we have used exhibit genomic and mutational profiles similar to those of HGSOC patient-derived organoids. Furthermore, in addition to our analysis of HGSOC copy number (CN) signatures on the three cell lines, we have conducted an additional exploration (**Reviewer Figure 2, new Supplementary Figure 4B**) using a pan-cancer study defining 17 CN signatures¹. Again, we observed that the cell lines exhibit profiles of signature activities capturing a diversity of CIN features that are also observed in human tumours. For these reasons, we think that our chosen cell lines, when added into the pentaculture model, represent genetic and biological characteristics of HGSOC (**Figure 4E**) and human malignant cells in general (**Supplementary Figure 4B**) whilst offering an accessible and practical in-vitro model platform (lines 254-259).

Reviewer Figure 2 (now Supplementary Figure 4B) – Stacked bar plot showing the abundance of seventeen copy number signatures across the three cell lines. Copy number signatures were defined according to ¹.

3. The authors conclude from bulk RNA sequencing that unsupervised clustering segregates samples according to malignant cell types rather than the donor of the normal cells, suggesting that malignant cells are the primary drivers (Figure 2B). However, this conclusion is not supported by this presented analysis. Bulk RNA-seq inherently masks cellular heterogeneity, making it insufficient to attribute clustering solely to malignant cell types. Moreover, what stands out more clearly is that the Pentacultures cluster together as a group, and not with their corresponding cancer cell types. If the authors aim to explore intercellular influences or the contribution of specific cell types within these cultures, single-cell RNA sequencing would be a more appropriate and informative approach.

We acknowledge that bulk RNAseq can mask cellular heterogeneity, making direct attribution of clustering solely to malignant cell types challenging without further deconvolution. In the present study, our primary focus was to introduce the pentaculture

system as a model incorporating macrophages – a novelty for *in vitro* multi-cellular systems. In our RNAseq experiment, by including individual cell types (malignant cells, fibroblasts, mesothelial cells and monocytes) alongside the pentaculture samples we were able to observe the enrichment of macrophage and microenvironmental transcriptional programs in the pentacultures compared to the individual cell type controls (**Figures 2C-D**) which we strengthened with further *in vitro* and *in silico* analysis in following figures. In **Figures 3 A-D**, we used our scRNAseq data from human HGSOC omental metastasis tissue biopsies where we had performed an in-depth characterisation of the macrophage population² to deconvolute and interpret the bulk RNAseq profiles of our models. These analyses revealed the presence of key macrophage subpopulations consistent with those identified in patient metastases within our pentacultures and the complexity of the macrophage population varied according to the malignant cell line used (as detailed in **Figure 3A** and **3C**). We believe this validation supports our bulk RNAseq findings albeit the effect of the individual stromal cells with a detailed intercellular communication analysis would need further exploration in a separate study. We include below a PCA analysis to enhance the information of the unsupervised clustering and illustrate the complexity of the variance in the data beyond the differences between the malignant cells used in the study (**Reviewer Figure 3**).

Reviewer Figure 3 - PCA plot illustrating sample clustering on the first two principal components and scree plot of the variance captured over ten principal components.

4. In Figure 2C, how can the authors conclude that differentiation into macrophages occurred in their culture based solely on GO term analysis? These GO terms are also enriched in monocytes, so this does not provide definitive evidence for macrophage differentiation.

We observed increased enrichment of several GO terms relating to macrophage differentiation compared to the individual cell-type samples as well as to monocytes. We have refined **Figure 2C** (now **Figure 2D** in the revised manuscript) to illustrate only GO terms with a significant change between pentacultures vs single cell controls (Welch t-test, Benjamini-Hochberg (BH) adjusted $p \leq 0.05$) and included a **Supplementary Figure 2A** (**Reviewer Figure 4**) identifying the pathways with increased expression compared to the monocytes control. Our intention with **Figure 2D** was not for the pathway analysis alone to serve as proof of differentiation. The pathway analysis in **Figure 2D** was designed to highlight pathways upregulated between malignant and primary cells in 2D and the pentacultures, complementing our evidence of macrophage differentiation shown in **Figures 1B, 1D, and 1F**. The enriched GOBP pathways in **Figure 2D**, particularly those directly associated with macrophage chemotaxis, migration, and differentiation, indicate an activated

and differentiated macrophage status within the pentacultures that is significantly distinct from isolated monocytes. We have now revised the manuscript to clarify the use of the GOBP pathway analysis in **Figure 2D** (lines 142-146).

Reviewer Figure 4 (new Supplementary Figure 2A) - Heatmap illustrating GSVA enrichment scores of macrophage-related Gene Ontology Biological processes (GOBPs). Welch t-test, BH adjusted $p \leq 0.05$ for penta-cultures vs single cell-types and mean enrichment scores greater than monocytes.

5. In Figure 2D, it is unclear which GO terms specifically support the conclusion of a tissue-like environment. The figure is poorly presented and does not convincingly support the authors' claims. Simply listing GO terms feels more like a supplementary figure. Higher-resolution data, with single-cell RNA sequencing, along with validation of key marker genes at the protein level, with appropriate controls, would be necessary to robustly substantiate the findings.

We appreciate the reviewer's critique of **Figure 2D** and the need for stronger evidence of a tissue-like environment. Although original G164 and OvCAR3 tumour sequencing is not available, we performed a direct correlation of GSVA enrichment scores for Hallmark pathways of interest using publicly available RNAseq data from the original AOCS1 tumour from which the cell line was derived. This analysis showed that the transcriptomes of each of the three AOCS1 pentacultures positively correlated with the original tumour's gene expression signature, whereas the AOCS1 cell line cultured alone exhibited a negative correlation (**Reviewer Figure 5, new Supplementary Figure 2B**). This finding further suggests that our pentaculture system induces a tumour microenvironment state that mimics the original human tumour. We have revised the manuscript to improve the clarity of these points (lines 148-154).

Reviewer Figure 5 (new Supplementary Figure 2B) - Correlation plots of GSVA enrichment scores of Hallmark pathways in AOCS1 tissue vs AOCS1 penta or vs the AOCS1 cell line in monoculture. Pearson's correlation coefficient and p -value are illustrated on the plots.

6. It is unclear whether the genomic and transcriptomic profiling of the cancer cell lines was performed on the isolated cell lines or within the context of the Pentaculture system. This distinction is important, as the microenvironment can significantly influence cancer gene expression. Additionally, this characterization of the cancer cell lines feels disconnected from the main narrative of the manuscript. The conclusions drawn regarding their relevance to macrophage states are underdeveloped and lack clear integration into the broader context of the study. Also, it is broadly known in many cancer and literature on this topics should be thoroughly reviewed by the authors (e.g. Weverwijk, De visser, nature review cancer 2023)

We appreciate the reviewer's comments on the profiling of our cancer cell lines and their integration into the manuscript. To clarify, the genomic and transcriptomic analyses were performed on the isolated malignant cell lines, i.e. as a single-cell type not incorporated into the pentacultures. This was done because our data (**Figures 2 and 3**) suggest that the intrinsic properties of the malignant cell lines are major drivers of the distinct microenvironments observed. We acknowledge the extensive literature on the role of cancer cells in shaping the TME, as highlighted by Weverwijk and De Visser. Our work's novelty lies in applying this established knowledge within a complex *in vitro* pentaculture model. For example, the above review described how higher STING pathway expression correlated with an inflammatory cytokine tumour microenvironment. In agreement with this, we found that the HRD-deficient G164 cell line has high STING pathway expression compared to the other two cell lines, and the highest level of inflammatory cytokine secretion **Figure 3J**. (see below **Reviewer Figure 6** and **new Supplementary Figure 4E**).

We have revised the manuscript text to better integrate this rationale and provide additional context regarding the established literature (lines 234-236 and 268-274).

Reviewer Figure 6 (new Supplementary Figure 4E) – Heatmap illustrating upregulation of STING pathways in G164 malignant cell compared to AOCs1 and OvCAR3 (one-way anova $p = 0.0005$ and $p = 0.003$ for REACTOME_STING_MEDIATED_INDUCION_OF_HOST_IMMUNE_RESPONSES and ISHIKAWA_STING_SIGNALING respectively).

7. Statistical analyses are missing from many figure panels, but their inclusion would be important to appropriately support and nuance the authors' claims.

The information about the statistical analyses has now been added to each figure legend.

8. Legends need more information

We have added more information to the figure legends e.g. Figure 1 legend now contains all the statistical tests used to calculate significance and it has been rephrased. We hope this makes it clearer and more comprehensive.

9. Fig 1 D axis labeling and legend not clear.

We hope that the changes are an improvement.

10. The graphical presentation in Fig. 1F is somewhat unclear. Were multiple donors analyzed? Additionally, could you include data for day 0 (monocytes) and day 14?

We appreciate the reviewer's feedback regarding **Figure 1F**. To clarify, the data for **Figure 1E** shows the D0 time point with three different monocyte donors. **Figure 1F** shows a representative plot of macrophage differentiation at day 3 and day 7 of one of the previous donor in a pentaculture experiment. **Figure 1D** summarises day 7 macrophage differentiation from five different monocyte donors in the pentacultures. Hence **Figure 1F** is an example of one of the experiments shown in **Figure 1D**. (see lines 111-117).

11. In Fig3, I-J, I am missing the monocytes as control.

The data in **Figure 3I-J** represent the secretome collected from the pentacultures or the malignant cell lines in monoculture, i.e., the active secretion of molecules into the conditioned medium over time. However, a direct secretome control of monocytes only is not technically feasible as monocytes in 2D monoculture don't survive without addition of exogenous growth factors. Hence, the monocytes are immediately introduced into the pentaculture system upon thawing, without *in vitro* culture. While we could quantify cytokine and chemokine levels in monocyte cell lysates, these intracellular measurements are distinct from secretome analysis.

12. Although the video quality is high, the objective and key observations are not clearly communicated. Adding annotations and labels would greatly enhance clarity by guiding the viewer to the relevant events. Specifically, it is not evident that CD47 blockade results in increased phagocytosis. If this is a key conclusion, more direct visual evidence of phagocytosis should be shown. Additionally, the significance of the observed wobbling motion is unclear, why should this be interpreted as an indicator of phagocytic activity? Clarification or supporting rationale would be helpful.

We appreciate the feedback regarding the clarity of our videos. To enhance clarity and guide the viewer, we have added annotations and labels to the videos. Specifically, we've included arrows pointing to effective interactions (readout of phagocytosis), providing more direct visual evidence of this process. However, our original manuscript provided quantitative measurement data to support our findings on phagocytosis. For both **Figure 5D** and **6G**, we measured the percentage of macrophages positive for red cancer cells and the effective distance between macrophages and the nearest cancer cells. We graphed macrophages located at less than 20µm from cancer cells, under the rationale that such close proximity or direct overlap indicates effective interaction and potential phagocytosis.

We also understand the request for clarification on the wobbling macrophages and its interpretation as an indicator of phagocytic activity. In our previous work³, we observed diverse movement patterns in both CD8 T cells and macrophages in *ex vivo* human and mouse tissues, though we couldn't provide a biological explanation for these variations at that time. In this work, we applied the same analysis to macrophage movement within our pentaculture model. We observed that the wobbling population was the only population whose proportion significantly increased together with the increase in phagocytosis. Based on this correlation, we hypothesise that this specific movement pattern might reflect a more phagocytic state. However, we want to be clear that we cannot definitively conclude that all the wobbling macrophages are phagocytic macrophages. Their wobbling nature could also be influenced by other factors within the microenvironment, such as the density of the extracellular matrix (ECM), the presence of other cell types, or localised depots of cytokines and chemokines that might restrict their movement.

We have now addressed these comments in both the Results and Discussion sections of the revised manuscript (lines 375-378 and 443-446)

13. The conclusion that increased phagocytic activity in post-chemotherapy samples from responders, compared to those from relapsed patients, implies that enhancing macrophage phagocytic capacity may improve chemotherapy response is not well supported. An alternative explanation could be that effective chemotherapy leads to greater tumor cell death and debris accumulation, which in turn triggers macrophage activation and phagocytic activity as a secondary effect, rather than a cause of therapeutic success. Could the authors clarify whether their data distinguish between these possibilities, and comment on the directionality of this relationship?

We thank the reviewer for this comment. We agree that both explanations are possible. Our current data cannot definitively distinguish between these two possibilities. We suggest that enhancing macrophage phagocytic activity is a potential mechanism that contributes to improved therapeutic responses, as it could potentially happen after chemotherapy. We have commented on this in the revised manuscript (lines 342-348).

Reviewer #2 (Remarks to the Author):

Malacrida et al. report the development and initial characterization of high grade serous ovarian cancer (HGSOC) “pentacultures”. This organotypic cell system includes cancer cells, mesothelial cells, fibroblasts, adipocytes and monocytes, the latter of which differentiate into macrophages in the culture. The adipocytes, mesothelial cells, and fibroblasts are obtained from omental biopsies and cultured *in vitro*. Three separate donors are used for this study, although the authors imply that they have studied many more (see below). The monocytes are purified from peripheral blood of a single donor. Using these pentacultures, the authors report several key findings. First, tumor cells seem to drive most of the macrophage polarization in the cultures, as shown by differences in transcriptomic and proteomic data, which had more effect than varying the origin (i.e. donors) of the other cell types. Second, each HGSOC cell line drives different degrees of macrophage polarization and different spatial architecture. Third, the authors report intriguing macrophage-directed effects on tumour cell viability, which could, at least for one cell line, be modulated with monoclonal antibodies targeting “do not eat me” signals.

Overall, this is an interesting, well written, and technically well-performed study that should be of general interest to the readership of Nature Communications. There are, however, some conceptual and technical issues that should be addressed in a revision.

Major Issues:

(1) A penta-culture (actually, a sexta-culture, as it includes endothelium) of the ovarian omentum, incorporating a macrophage cell line, has been described previously (Estermann et al., *Biomaterials*, 2023), albeit that work was more focused on dissemination of HGSC within the omentum than on the issues discussed here. In some ways, the authors of the current paper have also reported a 5-cell type culture (see Malacrida et al. 2021, Figure 4F), although that work focused on platelets, not macrophages. The authors should discuss the similarities and differences between their system and the previous reports. Also, they should emphasize the novelty of this paper, which is the demonstration that tumour cells drive the TME (at least those components of the TME that are present in this culture), and the “don’t eat me signal” aspects. The utility of their system in enabling the study of monocyte-> macrophage differentiation should also be discussed.

We appreciate the reviewer’s positive comments. We are aware of the contributions from other multicellular *in vitro* models, such as Estermann et al.⁴, as well as our own prior work⁵. We agree that it is important to discuss how our system is similar to and different from these previous reports, and to emphasise what makes our paper novel.

Regarding Estermann et al.’s work, it represents an important step forward in mimicking the omental microenvironment. While both their model and ours aims to capture the complex cellular interactions in the ovarian cancer omentum, the critical difference comes down to the origin of the cells. Estermann et al used immortalised cell lines to create the microenvironment components. Notably, they used the acute monocytic leukemia cell line THP-1 as a source of monocytic cells. In contrast, we build the microenvironment of our current pentacultures, (and our previously published models with fewer cell types), using primary normal human cells isolated directly from uninvolved omentum from patients and human peripheral blood monocytes. Successfully differentiating macrophages from human peripheral blood monocytes in a multi-cellular model has been one of the challenges in the field of 3D models. As far as we are aware, there is also currently no ovarian organoid model with successful incorporation of macrophages. As for our own previous work⁵, we focused on the early stages of metastasis, particularly on how cancer cells stick and invade an omental

structure comprising adipocytes, mesothelial cells and fibroblasts in the presence of platelets. Our current paper focuses on later-stage metastatic mechanisms, specifically on the role of macrophages. The overall design and the inclusion of five cell types is a progression in our efforts to build more relevant *in vitro* models, and we are asking a different question. We focused on macrophages because they are a dominant immune cell in HGSOC omental metastases, and they are increasingly seen as one of key reasons why immunotherapy does not work in HGSOC treatment. We also explored mechanisms of targeted therapy resistance by looking at the "do-not-eat-me" signals, especially CD47. This shows our model can be used to test therapies aimed at overcoming these resistance pathways. Finally, we characterised the relevance of our model using existing scRNAseq data from human HGSOC biopsies to confirm that the macrophage populations we see in our *in vitro* system are similar to those found in patients. We expanded on these details in the revised manuscript (lines 72-75, 84, 412-415, 455-465).

(2) Surprisingly, the authors do not comment on whether the tumor cells also drive fibroblast differentiation to CAFs (or induce additional fibroblast/CAF differentiation from mesothelium). Alternatively, does culturing the primary omental fibroblasts *in vitro* prior to their incorporation into pentacultures induce activation similar to that of CAFs? Several reports in other tumor systems have emphasized the role of CAFs in directing macrophage polarization, so this point would seem to be important.

We appreciate the reviewer's comment regarding fibroblast differentiation. While our primary focus in this study was cancer cell-macrophage interactions, we agree that other cell types, especially fibroblasts, might contribute and might also be influenced by the malignant cells. Our preliminary results indicate that malignant cells might also drive fibroblast activation and differentiation transcriptional programmes (**Reviewer Figure 7, new Supplementary Figure 3G**) as well as distinct fibroblast phenotypes within our model, similar to our observations with macrophages, though less strongly. Using immunofluorescence staining, we found that α SMA expression was significantly higher in AOCS1 and G164 pentacultures compared to the OvCAR3 pentacultures (**Reviewer Figure 8, new Supplementary Figure 3H**). However, the expression of FAP (fibroblast activation protein) on the fibroblasts and mesothelial cells in the pentacultures measured using flow cytometry, showed no significant difference in the pentacultures with the three malignant cell lines (**Reviewer Figure 9, new Supplementary Figure 3I**).

We have incorporated this information into the revised manuscript (lines 216-225). We feel that any further analysis is outside the scope this paper, but we will be following this up in the future.

Reviewer Figure 7 (new Supplementary Figure 3G) - Heatmap of Fibroblast GOBPs with increased enrichment in penta-cultures compared to single cell-type monoculture controls (Welch t-test, BH adjusted $p \leq 0.05$)

Reviewer Figure 8 (now Supplementary Figure 3H) – Representative maximum-intensity projections of Z-stack images from pentaculture models. DAPI (cyan), cancer cells (red), and α SMA-positive fibroblasts (magenta). α SMA fluorescence intensity was normalized to staining area. Data represent mean \pm SEM ($n = 3$), Welch's unpaired t -test.

Reviewer Figure 9 (now Supplementary Figure 3I) – Percentage of FAP⁺ fibroblasts and mesothelial cells in pentaculture cultures after 7 days, quantified by flow cytometry. Data presented as mean \pm SEM ($n = 3$).

(3) The authors state that they used 3 donors for the omentum-derived cells for this study, but in the Discussion (line 366), they say there are >50 donors for the first three cell types and 5 for the monocytes. To what extent have these other donors been characterized in this system? Also, what types of surgeries were the patients who donated omentum undergoing (i.e., what were their underlying disorders)?

We thank the reviewer for this comment and have now clarified this in the revised manuscript. We used three different donors for omentum-derived primary cells in the bulk RNA-sequencing experiment in **Figure 2**. All other experiments in the study have included multiple primary cell donors, as reflected by the ">50" and "5" stated in the Discussion. Regarding patient characteristics, primary adipocytes, fibroblasts, and mesothelial cells were isolated from macroscopically uninvolved omentum from gynaecology cancer patients undergoing debulking surgery, including both neoadjuvant chemotherapy (NACT)-treated and chemotherapy-naïve individuals. This detail has been clarified in the revised manuscript (lines 126-128, 484-485).

(4) In Figure 1, the authors indicate that monocytes spontaneously differentiate into macrophages when introduced into the quad-culture model. An important missing control would seem to be incubating the monocytes without any cancer cells; have they done this? Also, even if the cancer cells are necessary to start the monocyte differentiation process, are they sufficient? Or is there cross talk with other cell types (e.g., CAFs, as per point 2)?

We appreciate the reviewer's comment on monocyte differentiation and the need for controls. We have now performed experiments addressing the necessity and sufficiency of

cancer cells. We cultured monocytes in a 3D model of adipocytes, mesothelial cells and fibroblasts without cancer cells (tetraculture). While fibroblasts and mesothelial cells were sufficient to initiate monocyte-to-macrophage differentiation (**Reviewer Figure 10** and **new Supplementary Figure 3B**), in these experiments the levels of macrophage markers and secreted proteins were different from the pentaculture results shown in **Figure 3E-H** (**Reviewer Figure 11** and **12** and **new Supplementary Figure 3C** and **3D**). This suggests that malignant cells are necessary for development of the different macrophage subsets. Also, in a more simplistic 3D spheroid model (cancer cells + monocytes plated in ultra-low attachment plates), monocytes began to acquire some phenotypic differences. However, the absence of cytokines, such as M-CSF and CCL2 which we believed are mainly secreted by fibroblasts and mesothelial cells (**Reviewer Figure 13** and **new Supplementary Figure 3F**), resulted in poor macrophage viability after four days. While stromal cells initiate differentiation and cancer cells drive subset specification, we believe that the synergistic cross talk within the complete pentaculture is essential for sustained viability and the full phenotypic diversity of macrophages. These results are also described in the revised manuscript (lines 192-204, 210-215).

Reviewer Figure 10 (now Supplementary Figure 3B) - Flow cytometry analysis of the tetraculture (pentaculture minus cancer cells) expression of CD45+ and CD163+, CD206+, and HLADR+ of viable CD45+ cells. Data presented as mean ± SEM (n=3).

Reviewer Figure 11 (now Supplementary Figure 3C) - Flow cytometry analysis of the tetraculture expression of CD169 and GPNMB macrophages alongside the relative expression in the different pentaculture (same graphs as Figure 3E and 3F). Data presented as mean ± SEM (n=3).

Reviewer Figure 12 (new Supplementary Figure 3D) - CCL4 and CXCL10 released by the tetra-culture alongside the pentaculture graphs from Figure 3G and 3H. Data presented as mean \pm SEM (n=5).

Reviewer Figure 13 (now Supplementary Figure 3F) - MCSF Elisa using the supernatant from malignant cells and primary cell types in monoculture. Data presented as mean \pm SEM (n=3 or 5 different primary cells donors). Ordinary one-way ANOVA.

(5) The authors perform the studies in this paper over 7 days. How long do the cultures last beyond that? Also, they comment (and show data that) the viability of the cancer cells is preserved over the 7-day period, but what about the other cell types?

The original submission, **Supplementary Figure 1A** (now **Supplementary Figure 1B**) showed the overall viability of the entire pentaculture, reflecting the viability of all cell types. We have now included viability for each cell type within the pentacultures in the revised manuscript (**Reviewer Figure 14** and **Supplementary Figure 1C**). Regarding culture duration, our pentacultures could be maintained for up to 14 days. However, at this time point we observed a significant decline in macrophage viability. Therefore, we did not test cultures beyond 14 days (see lines 99-104).

Reviewer Figure 14 (now Supplementary Figure 1C) - Viability of the different cell populations at day 7. Data shown as mean \pm SEM (n=3).

(6) They note that a major limitation of the current work is the absence of adaptive immune cells (I would include other innate immune cells, e.g., gMDSC/neutrophils, NK cells). A priori, it would seem that at least some of these could be added. Could the authors comment on whether they have tried to do so in the Discussion. Also, is there a reason why they chose to use immortalized HGSOc lines instead of primary tumor cells or organoids?

We appreciate this comment from the reviewer. We completely agree that more immune cells would enhance the relevance of our model. While our current study focused on cancer cell-macrophage interactions, we are actively working to increase the complexity of our models. One ongoing project in our lab is integrating CAR T cells in the pentacultures, but we believe this is outside the scope of the current paper. Regarding the use of immortalised HGSOc cell lines, we apologise to the reviewer for any confusion. Please also refer to our response to **reviewer 1 - point 2**. AOCs1 and G164 were established from patient samples and self-immortalised in culture. Although OvCAR3 was purchased from ATCC, all three cell lines were extensively characterised at proteomic, transcriptomic and genomic level. As demonstrated in **Figure 4E**, these cell lines exhibit genomic and mutational profiles that are similar to those of HGSOc patient-derived organoids and HGSOc biopsies.

We have added a comment regarding the future directions of our models to the Discussion section of the revised manuscript (lines 455-460).

(7) Line 151-154: the authors state that CIBERSORTx ..revealed that fibroblast and mesothelial cells were present at similar abundance..”, but to my eyes, it looks like there are significantly more of these cells in AOCs1 cultures—am I wrong?

We appreciate the reviewer's observation regarding **Figure 3A**. We checked the results, and while the graph might visually suggest slight differences, we confirm there is no significant difference in the abundance of fibroblasts and mesothelial cells across the various pentaculture models, including AOCs1 (**Reviewer Figure 15**).

Reviewer Figure 15 – Percentage abundance of fibroblasts and mesothelial cells from CIBERSORTx analysis (Figure 3A in the manuscript). No statistical difference found between the different pentacultures. One-Way ANOVA and Welch's *t* test.

Moreover, as CIBERSORTx only provides indicative proportions based on RNA expression, we have now provided flow cytometry quantification for each population in **Supplementary Figure 1A** (see also **Reviewer 1, point 1**).

(8) Lines 159-60, the authors correctly note that the percentage of G164 cells is significantly higher by flow, as predicted by CIBERSORTx. But why are AOCS1 and OVCAR1 identical when they are significantly different in the CIBERSORTx analysis?

We appreciate the reviewer's comment regarding the CIBERSORTx and flow cytometry data for AOCS1 and OvCAR3 abundances. While CIBERSORTx is a robust way to deconvolute our bulk RNAseq data from the *in vitro* models using scRNAseq derived signatures from HGSOc biopsies, some gene expression signatures might overlap across cell populations and could lead to discrepancies between *in silico* predictions and actual measurements. For this reason, some of our deconvolution results, including cancer cell abundance and macrophage clusters, were validated at protein level using flow cytometry or ELISA (see **Figure 3E-H**).

(9) Line 197, Why don't the authors also comment on CDK12 loss in this line? And what about CCNE1 amplification in OVCAR3?

We thank the reviewer for this comment. We agree that a discussion of these specific genetic alterations is important for providing a more complete context for the cell line characteristics.

CDK12 loss is well-documented for its association with defects in DNA repair, which is consistent with the HRD-deficient profile of the G164 cell line. This genetic signature also aligns with results from a previous publication⁶ showing higher sensitivity of this cell line to PARP inhibitors when compared to AOCS1. Regarding CCNE1 amplification, a common alteration in HGSOc, we observed that while all three cell lines show this amplification (**Figure 4C**), it is present at the highest levels in OvCAR3 and the lowest in G164. This finding correlates with higher CIN signature 4 activity in OvCAR3 and AOCS1 and was also in line with whole genome duplication events observed in these lines, **Figure 4E**. We have incorporated these details into the revised manuscript (lines 240 and 249-251).

(10) Lines 311-313, referring to the G164 pentacultures, the authors state that macrophages resided mainly in the stroma. But from the micrograph in 6D, it looks mainly like there are far fewer macrophages, period. Could they comment?

We recognise that a single image, such as the one presented in **Figure 6D**, can sometimes lead to biased visual observation of cell abundance due to the heterogeneous spatial distribution within a 3D culture model. Our primary goal for that specific image was to capture a representative area proving the predominant stromal localisation of the macrophages in the G164 pentaculture. As shown in new **Supplementary Figures 1A and 1C**, we have now included detailed analyses presenting both the proportions of different cell types and the viability of each individual cell compartment at the end of the 7 days of culture for each pentaculture model. These quantitative data, along with the results in **Figure 1C**, show that the overall percentage of macrophages is comparable across the different pentaculture models.

(11) The quantification in Fig. 5D does not agree with the photo in 5C, at least to my eyes. We appreciate the reviewer's observation regarding the visual discrepancy between **Figure 5C** and its quantification in **Figure 5D**. As previously mentioned, static images are illustrative and may not fully capture the dynamic 3D interactions that we see in the videos. However, quantification in **Figure 5D** is derived from the average of 5 different time frames within the 30-minute videos. This approach accounts for macrophage movement and heterogeneous localisation.

Minor issues:

- 1) Line 39, HGSOC is misspelled; Line 306 "the" should be capitalized.
- 2) In Fig 1A, I found it difficult to see the fibroblasts in the cartoon. I suggest using a different color for more contrast.
- 3) Lines 104-106; the authors state that "...pentacultures retained a substantial population of monocytes, as indicated by CD45+ staining..." But macrophages are also CD45+? How can this statement be used to support the retention of MONOCYTES?

Minor Issues 1, 2 and 3 have now been addressed in the revised manuscript.

- 4) Figs. 2C-E are too small to see without magnification (i.e., when printed out at 1x). So are Fig. 4D and 6A and Supp Figs 2 and 3 (especially those numbers in blue boxes).
- 5) Figure 1F. To facilitate reader interpretation, I recommend adding arrows in the graph that indicate the change of the center of mass of each population between days 3 and 7

We have now improved the quality of the figures mentioned in Minor Issues 4) and 5).

- 6) Why aren't all the chemokines and cytokines presented in the same order in Figure 3I for pentaculture and 2D cultures?
- 7) Line 205, CIN should be defined as chromosomal instability when used for the first time.
- 8) Lines 212-214 is worded awkwardly. Also, flow cytometry does not measure "expression"; it measures "surface expression" (e.g., Fig 1 legend, and elsewhere in the paper"

We have now made the changes suggested.

Reviewer #3 (Remarks to the Author):

The authors present an interesting preclinical model for high-grade serous ovarian cancer, incorporating tumor cells, primary fibroblasts, mesothelial cells, adipocytes, and monocytes using three previously established cell lines.

This manuscript builds on their previous work on “tetracultures”, further incorporating monocytes from healthy donors to their TME model. They show that the monocytes differentiated into macrophages, without exogenous cytokines, in the pentacultures. The authors then describe the model using IHC, flow cytometry RNAsequencing and whole genome sequencing. They highlighted the potential influence of malignant cell genomic, transcriptomic and proteomic heterogeneity on the TME. The findings on the functional characterization are interesting. The work is highly relevant and timely; however, the strength of the conclusions and translational impact would be improved with more detailed characterisation of the models and source tumors (see also **Reviewer 1 – point 5**).

1. For the representativeness of the models: Fig 1 B and S Fig 1b: is the human biopsy from the same patient? It is difficult to assess the cell morphology. To conclude how well the models match the human tumors, it would be important to should show

- 1) matching human tumor and model H&E stainings

While we appreciate this point, due to our model setup, which combines cancer cell lines with primary stromal and immune cells from multiple patients, direct patient-matched histological comparisons are not feasible. However, the strength of our model lies in its ability to replicate key characteristics observed in the human omental metastatic microenvironment.

- 2) higher magnification of H&E to allow for assessing the cell morphology

We have now provided these images

- 3) It is difficult to see any tumor cells in the models - the authors should show also tumor marker stainings e.g. CK7, PA8 in the models. Based on Fig1 B and S Fig 1B, HLA-DR is widely expressed and most of the cells seem to be macrophages and tumor cells cannot be identified from the figures.

See point below

2. Row 101: “AOCS1 and OvCAR3 cancer cells were located close to the stromal cells in the pentacultures, while G164 cancer cells generally formed clusters away from the stromal cells.” Where is this shown? Is this quantified? Deeper characterisation of the full composition (see point 1) would also better support the conclusion that can be made on the influence of the malignant cells on the morphology and composition of the models.

To respond to the reviewer’s points, we are providing here in **Reviewer Figure 16** (now **Supplementary Figures 1E and 1F**) images of our different pentacultures. We believe that these immunofluorescence images demonstrate the differences: AOCS1 and OvCAR3 malignant cells (red) appear integrated with α SMA⁺ fibroblasts (magenta – top panel) or CD14⁺ macrophages (green – bottom panel), whereas G164 cells form distinct aggregates surrounded by the stromal components. These observations are also supported by **Supplementary Videos 1–3** and **Figure 1B** (CD45 panel), which show that macrophages localize closer to cancer cells in the AOCS1 and OvCAR3 pentacultures, while in the G164 pentaculture, macrophages predominantly reside within the CD90⁺ stromal regions (fibroblasts and mesothelial cells).

Reviewer Figure 16 – Immunofluorescence staining showing the different morphologies of the pentaculture models. Representative maximum intensity projection images of the different pentacultures. Top panel: DAPI in cyan, cancer cells in red, αSMA in magenta, scale bar 20µm. Bottom panel: CD14⁺ macrophages, cancer cells in red, CD90⁺ stroma in magenta, scale bar 50µm.

3. For the RNAseq and analysis: Fig 2 a - it is difficult to assess which conditions were sequenced and compared. This figure could be improved to better show the different conditions and comparisons. Figure 2 b is not really informative as part of the results, but rather shows sample-specific clustering. Was there a batch effect? Was that corrected for?

We thank the reviewer for this comment. All samples were processed under identical conditions, including collection, instrument, and run time, so there is no technical batch effect. The experimental variables in the setup were the type of malignant cells, the type of stromal cells, and the specific patient donor of the stromal cells. However, the observed clustering was primarily driven by the type of malignant cells used in the cultures.

4. Tissue composition: bulkRNAseq deconvolution has known biases for decomposing the signal - the authors should validate these findings using another technology, for example, immunofluorescence or flow cytometry.

We appreciate the comment made by reviewer and we agree. We have indeed validated our findings using another assay: **Figure 3B** presents validation of cellular proportions in the pentacultures via flow cytometry, and **Figures 3E-H** demonstrate validation of specific macrophages using both flow cytometry and ELISA. Also, the differences in cellular proportions can now be also appreciated from **Supplementary Figure 1A** (see response to Reviewer 1, point 1 above).

5. Fig 3: the tumor cell fractions were quite low after - could this be corrected using higher numbers of cancer cells when establishing the cultures? Did the authors test different ratios of the cells in the setup of the pentacultures?

We did test different initial cancer cell numbers and ratios. While altering these inputs changed the absolute number of cancer cells present at the end of the culture, it did not affect the relative differences in phagocytosis observed between the malignant cell lines. OvCAR3 and AOCS1 consistently remained significantly more phagocytosed by the macrophages than G164.

6. How much of the transcriptional differences of the pentacultures stem from the significantly different gene expression profiles of the tumor cells? To show that the differences in the immune/stromal cell phenotypes are not just reflections of the (often poorly) tumor signal - scRNAseq (or bulk from flow-sorted populations) from the pentacultures would significantly strengthen the conclusions.

To identify genes that significantly changed between pentacultures, but not between the malignant cell monocultures, we explored the differential expression data for genes with a significant change in at least one of the penta-cultures vs the other pentaculture (BH adjusted $p < 0.05$ and $\log_{2}FC > |0.5|$) AND a non-significant change in any of the malignant cell monoculture pairwise contrasts (BH adjusted $p > 0.1$). This resulted in 244 genes which are illustrated in the heatmap below and indicate transcriptional differences between pentacultures beyond the basal malignant cell line differences. These genes reflected both additional changes in malignant cells as well as in the stromal compartment when in pentaculture. Of particular interest was cluster 1 containing PRGFRB, CCR2, TMEM132B, CD83, S100A11 and DCSTAMP which, we suggest, reflects the stromal compartment of the penta-cultures. The number of genes in the clusters was not sufficient for Gene Ontology and reactome pathway enrichment analysis. To inspect differences at the level of GO processes and reactome pathways, in a similar manner to gene level analysis, we explored gsva enrichment scores with a significant change between pentacultures (nominal $p < 0.05$) but not between the malignant cell monocultures (nominal $p > 0.1$). We noticed that the pathways that differed between pentacultures but not between monocultures were diverse, reflecting both further malignant cells changes such as axon guidance, metabolism and morphogenesis, but also the stromal compartment of the penta-cultures (defence response to tumour cells, regulation of fibroblast proliferation and T cell homeostasis)(**Reviewer Figure 17 – panel A is also new Figure 2E**).

Reviewer Figure 17 (panel A is also new Figure 2E) – Heatmaps illustrating (A) normalised \log_2 RPKM gene expression of differentially expressed genes with a significant change in at least one of the pentacultures vs the other pentacultures (BH adjusted $p < 0.05$ and $\log_{2}FC > |0.5|$) AND a non-significant change in any of the malignant cell monoculture pairwise contrasts (BH adjusted $p > 0.1$) clustered with k-means clustering; (B) GSEA enrichment scores of MsigDB GO and REACTOME collections with a significant change between pentacultures (nominal $p < 0.05$) but not between the malignant cell monocultures (nominal $p > 0.1$), clustered with k-means clustering.

7. Colour annotations are missing in Figure 4 B. The value of WGS exploration in only three tumor samples remains limited. Is G164 a functionally HRD tumor by e.g. functional RAD51 staining? Exploring more models with HRD vs HRP tumors would be really interesting!

The colour annotation for **Figure 4B** was originally placed below **Figure 4A**. We have moved it closer to **Figure 4B** to make it clearer. We agree that whole-genome sequencing on only three cell lines provides a limited view of genetic diversity. The purpose of this analysis was to establish a link between specific genomic profiles of our cell lines and their resulting TME phenotypes. We agree that exploring a greater number of models with diverse HRD/HRP status would be an important future direction. Regarding the functional HRD status of the G164 cell line, published work by Tamura et al.⁶ reports its higher sensitivity to PARP inhibitors when compared to AOC1.

8. In the end, it remains unclear how well do the observed models match the clinical tumors in terms of composition and phenotype. It would strengthen the conclusion if the authors had the original tumor samples, and were able to better show that they recapitulate the source

tumors. Further, the conclusion that the tumor genotypes shape the TME would need significantly more models to be established to reach statistical conclusions. E.g. the clinical relevance of these phenotypes could then be expanded to other public clinical datasets for increased impact.

We appreciate the reviewer's comment regarding the clinical relevance of our models in terms of composition and phenotype. Regarding the direct comparison with matched clinical tumours, it is technically impossible within the framework of our model. Our primary cells are isolated from macroscopically uninvolved omentum from patients. However, although original G164 and OvCAR3 tumour sequencing is not available, we performed a direct correlation of GSVA enrichment scores for Hallmark pathways of interest (related to the TME) using publicly available RNAseq data from the original AOCS1 tumour (see Reviewer 1, point 5 – **Reviewer Figure 5, new Supplementary Figure 2B**). Furthermore, we utilised scRNAseq signatures derived from human omental biopsies to deconvolute our data, confirming the presence of clinically relevant macrophage subsets. We further validated the abundance and functional states of these cells using flow cytometry and ELISA (see **Figure 3**). We have revised the manuscript to improve the clarity of these points (lines 146-155).

9. Figures 6 G-J only show two datapoints per condition and the statistical comparisons - please add biological replicates to these experiments.

We appreciate the reviewer's comment regarding the number of replicates in **Figures 6G-J**. We acknowledge that more biological replicates would increase statistical power. However, each data point is the analysis from a single video representing the average measurement of 40-50 macrophages over multiple timeframes. For this reason, we are now providing the results of the two experiments separately, one in the main **Figure 6G** and the second experiment in **Supplementary Figure 6F** (see below **Reviewer Figure 18**). Moreover, this analysis only serves as second confirmation of the results provided in **Figure 6E**. Given the internal consistency of each replicate, we believe these data points provide sufficient evidence to support our conclusions.

Reviewer Figure 18 (now Figure 6G and Supplementary Figure 6F) - Average percentage of macrophages at a distance less than 20µm from malignant cells. Data presented as mean ± SEM of 5 frames. The independent experiments have been separated.

References

- 1 Drews, R. M. *et al.* A pan-cancer compendium of chromosomal instability. *Nature* **606**, 976-983 (2022). <https://doi.org:10.1038/s41586-022-04789-9>
- 2 Elorbany, S. *et al.* Immunotherapy that improves response to chemotherapy in high-grade serous ovarian cancer. *Nat Commun* **15**, 10144 (2024). <https://doi.org:10.1038/s41467-024-54295-x>
- 3 Laforêts, F. *et al.* Semi-supervised analysis of myeloid and T cell behavior in ex vivo ovarian tumor slices reveals changes in cell motility after treatments. *iScience* **26**, 106514 (2023). <https://doi.org:10.1016/j.isci.2023.106514>
- 4 Estermann, M. *et al.* A 3D multi-cellular tissue model of the human omentum to study the formation of ovarian cancer metastasis. *Biomaterials* **294**, 121996 (2023). <https://doi.org:10.1016/j.biomaterials.2023.121996>
- 5 Malacrida, B. *et al.* A human multi-cellular model shows how platelets drive production of diseased extracellular matrix and tissue invasion. *iScience* (2021).
- 6 Tamura, N. *et al.* Specific Mechanisms of Chromosomal Instability Indicate Therapeutic Sensitivities in High-Grade Serous Ovarian Carcinoma. *Cancer research* **80**, 4946-4959 (2020). <https://doi.org:10.1158/0008-5472.Can-19-0852>

Point-by-point response to Reviewer 2

Reviewer #2 (Remarks to the Author):

This is a revised version of a manuscript that I reviewed earlier. The authors have made a good faith effort to respond to my queries and suggestions as well as those of the other Reviewers. In my opinion, this paper is quite important and warrants publication. However, I do have a few remaining issues/concerns that should be addressed before publication (not requiring re-review).

1) Most importantly, in my previous review, I asked about the source of the adipocytes, mesothelial cells, and fibroblasts used for the pentacultures. The text of the original manuscript (and this version) states only that they were obtained from patients undergoing “gynecological surgeries.” However, in their Response to the Reviewer’s Comments’, they state that these samples were actually from “uninvolved regions” of omentum in patients undergoing surgery for HGSOE. Given that these cells are typically bathed in cancer-associated ascites, and some have received neoadjuvant chemotherapy (!), it is highly likely, that they are not normal. The authors should discuss this possibility and its implications for interpreting their results. Also, I think that they should explicitly mention the source of these cells in the text on p. 4 and indicate whether the various donors used had received neoadjuvant chemotherapy or not.

We agree with the reviewer and have updated the manuscript to clarify the tissue sources and donor treatment status. We acknowledge that stromal cells from “uninvolved” omental regions may not be “normal” due to exposure to cancer-associated ascites or neoadjuvant chemotherapy, which could potentially pre-activate these cells. However, it is important to note that the results of our experiments remained consistent across biological replicates when using primary cells isolated from both chemo-naïve and post-chemotherapy patients. However, to ensure consistent transcriptomic result and avoid any bias, the bulk RNAseq experiment was specifically designed using primary cells from three chemo-naïve HGSOE patients. We have integrated these details and a discussion of their implications for our findings throughout the revised manuscript to ensure full transparency regarding our cell sources.

2) The flow plots in Fig. 1F should be described better—what is the y-axis, for example? Also, why not show the actual values for each population in a bar graph/dot plot or table?

We have now clarified the axes in Figure 1F, specifying that the plots represent marker expression versus side scatter area. We displayed these flow plots as we wanted a visual representation of the shift in macrophage markers from day 3 to day 7 to demonstrate how these cells differentiate over time. Moreover, the day 7 data shown are taken from one the experiments shown in Figure 1D. We have updated the manuscript and the figure legend to provide a more thorough explanation of these plots and enhance clarity.

3) In Fig 3B, rather than stating the y-axis measures “% viable cancer cells”, it would be better if they stated the exact measurement made either in the figure proper or the legend.

We appreciate the reviewer's suggestion to clarify the measurement in Figure 3B. However, the data represent the percentage of viable cancer cells as quantified by flow cytometry. This normalization was necessary to ensure comparison across different experimental conditions, accounting for variations in absolute cell counts between replicates.

4) The color code in Fig. 4B should be explained.

Colour code explanation has now been added to the figure legend.

5) The text of the second paragraph of p. 8 needs editing. It should be "...such as fibroblasts", and in the next sentence, "...and shows why external supplementation.."

This has now been corrected.

6) The reference to Adzibolosu et al. in the legend to Fig. 5 should be number 40, not number 13.

This is now corrected.